# Evaluation of Chemical and Nutritional Characteristics of Ricotta Cheese from Two Different Breeds: The Endangered Italian Teramana Goat and the Cosmopolitan Saneen Goat

**DOI:** 10.3390/foods13081239

**Published:** 2024-04-18

**Authors:** Marco Florio, Costanza Cimini, Francesca Bennato, Andrea Ianni, Lisa Grotta, Giuseppe Martino

**Affiliations:** Department of Biosciences and Technology for Food, Agriculture and Environment, University of Teramo, 64100 Teramo, Italy; mflorio@unite.it (M.F.); ccimini@unite.it (C.C.); fbennato@unite.it (F.B.); aianni@unite.it (A.I.); lgrotta@unite.it (L.G.)

**Keywords:** biodiversity, cheese quality, fermented dairy products

## Abstract

The present study aimed to compare the qualitative features of ricotta cheese produced by Teramana goats and Saanen goats raised in similar breeding systems and environmental conditions. The analyses were performed on ricotta after 0 (T0) and 5 (T5) days of storage at 4 °C. Ricotta cheese samples were subjected to chemical and physical analyses. The Teramana goat ricotta cheese was found to have a high-fat content characterized by a marked percentage of conjugates of linoleic acid (CLA). The reduction inketones and carboxylic acid revealed that Teramana goat ricotta cheese had greater oxidative stability during storage. According to the physical analyses, there are no differences between the two breeds in terms of color characteristics. Our findings underscore the importance of advocating for indigenous breeds, as evidenced by the compelling results observed in the production of ricotta cheese from Teramana goats.

## 1. Introduction

Over the last few decades, many local goat populations have declined dramatically in numbers [1]. Several factors have contributed to this decline: for instance, economic factors play an important role, indigenous breeds are typically less productive than the high-production cosmopolitan breeds, and this, combined with the increasing demand for animal products, has caused the number of animals in indigenous populations to decline.

Among these endangered native breeds, there is the Teramana goat, a breed that originated in the province of Teramo, in the Abruzzo region, in Italy. Teramana goat is medium-sized with a dark coat (primarily black or dark brown), sometimes exhibiting white stripes on the head. The head is long and has a straight frontal–nasal profile, and both sexes may have horns [2].

To safeguard these breeds, one strategy is to improve their value first, which can be achieved by enhancing breed-specific products. Indeed, consumer interest and demand for high-quality food have grown due to the increased interest in preserving health and the environment [3].

The initiatives will promote the on-site preservation of indigenous breeds among breeders. For instance, meat products from native swine breeds from the Abruzzo region were found to have a higher content of coenzyme Q10, a compound with high antioxidant potential, and better oxidative stability in stored cooked meat samples compared to those from hybrid pigs [4].

For this reason, in the previous work, we focused on the characterization of cheese produced from Teramana goat’s milk [5]. The milk and cheese were found to be rich in CLA, the consumption of which is frequently correlated with significant health benefits for consumers. Additionally, reduced concentrations of compounds, such as ketones and esters, and higher concentrations of carboxylic acids were observed, which would improve the oxidative stability of the cheese during the ripening process. These variations in volatile profile, lipolytic action, and technological characteristics were positively confirmed through sensory analysis. Indeed, the hardness, saltiness, and odor intensity of the cheeses increased, potentially leading to improved consumer acceptance of the product.

In this study, we focused on another product, ricotta cheese, which is an older and better dairy product obtained from whey [6]. It is extensively manufactured in the Mediterranean area. Ricotta is generally produced with sheep’s, goat’s, cow’s, or buffalo’s milk whey left after cheesemaking [7].

Cheese whey is a secondary product of the dairy industry, which still retains numerous valuable nutrients such as lactose, soluble proteins, minerals, and milk fat, and the preparation of ricotta cheese is one of the methods that enhance these by-products [8]. High residual sugar concentration, high moisture content, soft texture, homogeneous yellowish-white color, and the absence of starting culture input during manufacture are the characteristics of ricotta cheese [9].

Ricotta cheese is made by heating the whey, which causes the whey proteins to denature and separate. To further enhance this process and promote the coagulation of whey proteins, organic acids and salts can also be added. The product that emerges on the surface is collected in plastic baskets with small apertures that allow the drainage of the liquid phase [10]. Among consumers, the demand for traditional and high-nutritional dairy products is continually growing, and ricotta, due to its characteristics, satisfies this demand.

In the present study, the hypothesis is that nutritionally, the milk and the products generated from Teramana milk are superior to those produced from the milk of a cosmopolitan breed, the Saanen goat. In fact, in our recent study [5], a lower presence of saturated fatty acids in milk and cheese was shown to be in favor of monounsaturated fatty acids and CLA (linoleic acid conjugates) features connected to consumer welfare [11]. We evaluated the characteristics of ricotta produced from milk derived from the Teramana goat, comparing them with that produced by the Saanen goat, a more common breed, in order to assess the impact of different breeds on the qualitative characteristics of ricotta cheese.

## 2. Materials and Methods

### 2.1. Experimental Design, Cheesemaking, and Sampling

Forty animals of both breeds were used for this study. All animals, at the time of sampling, were homogenous in terms of body weight and stage of lactation. Additionally, all feed data were gathered by completing questionnaires by the farmers. Farmers were questioned about the type of feed they used, including the pasture composition and the type of feed used in the barn. In all the farms, the diet components were very comparable, given Abruzzo’s small reality [5].

To make ricotta, the whey from a previous cheese-making process that resulted in rennet breakage was retrieved, put into boilers, and heated to 87–89 °C without the use of exogenous acidification. At this temperature, small flakes appeared on the surface. Once the clots had formed, they were removed and put into molds that gave ricotta cheese its well-known shape. For each group, six ricotta cheeses of about 450–500 g were prepared. The yield of ricotta was around 9–10% for both groups. Three ricotta cheeses were stored immediately after manufacturing (T0), while the other three were left at +4 °C for 5 days (T5) and then subjected to sampling. All samples not immediately analyzed were collected in a vacuum and stored at −20 °C until the analysis.

### 2.2. Chemical Analysis and Color Measurement of Ricotta

The moisture of T0 and T5 ricotta cheeses was determined according to the AOAC methods [12]. The evaluation of total lipids in ricotta was performed by following the procedure reported by Innosa et al. [13], and the amount of total fat was expressed as a mean percentage on a dry matter (DM) basis.

The chromatic coordinates L* (lightness), a* (redness), and b* (yellowness) of ricotta cheeses were calculated using a CR-5 colorimeter (Minolta, New York, NY, USA). Each measurement was made using optical equipment with an aperture size that was set to 3 mm, and the sample was placed onto a glass Petri dish (33 mm) for reflectance. From the already measured parameters, the total difference in color (ΔE*ab) and the Yellow Index (YI) were calculated by using the formulas listed below:ΔE*ab = [(ΔCIE L*)2 + (ΔCIE a*)2 + (ΔCIE b*)2]1/2
YI = 142.86X (CIE b*/CIE L*)

### 2.3. Fatty Acid Profile of Ricotta

In order to induce the transmethylation of fatty acids, 500 µL of sodium methoxide in methanol (1:1, *v*/*v*) was added to the 70 mg of the ricotta T0 lipid that was extracted as described in the previous paragraph (chemical analysis and color measurement of ricotta). Then, according to the procedure described by Bennato et al. [14], fatty acid methyl esters (FAME) were separated. The ChromeCard Software (www.thermofisher.com, Thermo Fisher Scientific, Waltham, MA, USA) was used to examine the peak regions of each FAME found, and the values associated with each fatty acid were expressed as a relative percentage of total fatty acids. The sum of monounsaturated fatty acids (MUFA), polyunsaturated fatty acids (PUFA), and saturated fatty acids (SFA) was calculated using the value of each fatty acid. Additionally, the desaturation indices (DI) for C14:0, C16:0, C18:0, and CLA were determined using the formula reported by Innosa et al. [13].

### 2.4. Extraction and Separation by Sodium Dodecyl Sulfate-Polyacrylamide Gel Electrophoresis (SDS-PAGE) of Whey Protein

The protein profile of ricotta samples at time points T0 and T5 was assessed using SDS-PAGE, following the protocol outlined by Laemmli [15]. For protein extraction, 10 g of ricotta sample was dissolved in 10 mL of H_2_O and incubated at 37 °C for 15 min. Subsequently, 1 mL of 5% (*v*/*v*) acetic acid was added, followed by the addition of 1 mL of 1 N sodium acetate after 10 min. The samples were then filtered, and 200 μL of 100% (*w*/*v*) trichloroacetic acid (TCA) was added to 1 mL of the filtered solution. The samples were stored at −20 °C for 20 min and then centrifuged at 4 °C for 20 min at 12,000× *g*. The resulting pellet was washed three times with 1 mL of cold acetone at 12,000× *g* for 10 min each, and the supernatant was carefully removed. The extracted proteins were then quantified using the Bradford method [16] and separated on 12% SDS-PAGE gel, as per the procedure reported by Florio et al. [17]. Densitometric analysis of the visualized bands was then performed by exploiting the ImageJ software (https://imagej.net/ij/, National Institutes of Health, Bethesda, MD, USA), and the bands’ density was expressed as a relative percentage of the total protein.

### 2.5. Ricotta Volatile Profile

Volatile compounds (VOCs) present in ricotta cheese after 5 days of storage were extracted using solid-phase microextraction (SPME) and separated via gas chromatography (GC Clarus 580; Perkin Elmer, Waltham, MA, USA) coupled with a mass spectrometer (SQ8S, Perkin Elmer). The GC was equipped with an Elite-5MS column (dimensions: 30 × 0.25 mm; film thickness: 0.25 µm; Perkin Elmer). In brief, 3 g of ricotta were transferred into vials containing 10 mL of a NaCl solution (360 g/L) and 10 μL of an internal standard (4-methyl-2-heptanone). VOC adsorption was conducted using a divinylbenzene–carboxen–polydimethylsiloxane SPME fiber (Supelco, Bellefonte, PA, USA) exposed for 1 h at 60 °C in the headspace. The extracted VOCs were identified using the Kovats retention index following thermal desorption into the GC/MS. Data for each compound were expressed as relative abundance relative to the total identified VOCs.

### 2.6. Statistical Analysis

Statistical data analysis was performed utilizing the JMP Pro 14 software (SAS Institute, Cary, NC, USA). ANOVA (Analysis of Variance) was employed to assess the effects of breed (R), conservation (T), and the interaction between breed and conservation (R × T). Analysis of the fatty acids profile and volatile profile focused solely on the impact of the breed. Sample means were compared using Tukey’s HSD test, with differences considered significant for *p* < 0.05. Data were presented as least square means ± pooled standard error of the mean (SEM).

## 3. Results and Discussion

Rustic breeds represent an important genetic resource for biodiversity [3]. These animals are reported to produce high-quality animal products, which may be due to their natural diet and local environmental influences that contribute to the organoleptic properties of the product, representing a healthy and sustainable alternative to industrial products. Ricotta is produced using whey, which would otherwise be discarded as waste. Furthermore, ricotta is a nutritious food due to its high-quality protein content, essential nutrients, and low-fat content.

### 3.1. Physical and Chemical Evaluations of Ricotta

The first step to examine the differences between the breeds was performed by conducting chemical analyses and measuring the color of the ricotta cheese.

Teramana ricotta T0 showed a higher DM percentage compared to Saanen ricotta T0 (*p* < 0.001), and higher lipid content (*p* < 0.001 for R, *p* < 0.05 for T) (Table 1) was observed between the two groups. The same trend was observed in ricotta T5.

Regarding the chromatic coordinates (Table 1), lightness (CIE L*), yellowness (CIE b*), and YI were not affected by the breed. Significant differences were detected in the time of conservation of Saanen sample T5. After 5 days of storage, lightness (*p* < 0.01) decreased in Saanen ricotta T5, while values for b* (*p* < 0.001) and YI (*p* < 0.001) increased.

Ricotta made from the whey of Teramana showed a higher total fat value (expressed as a percentage of dry matter) compared to Saanen samples. As reported by Pizzillo et al. [6], the amount of lipids was higher in ricotta cheese made from the whey of the local breed.

Lightness (L*), yellow-blue coordinate (b*), and yellow index (YI) are the commonly used color parameters to describe color variability in dairy products. Color is an important factor that affects consumer acceptance, as the color of the food product is often associated with its quality and freshness. Regarding lightness (L*), yellowness (b*), and YI, no significant differences were detected between the two breeds at T0 and T5.

On the contrary, it was found that the color coordinate L* decreases significantly in T5 samples compared to T0 in Saanen ricotta cheese; the lightness L* of a dairy product can be directly connected to the moisture content [18], which decreases in Saanen ricotta samples at T5 compared to T0.

The Saanen ricotta samples showed significantly higher values of the color coordinate b* after 5 days of storage, indicating the propensity of the dairy product to acquire a darker appearance overall; the darkening of products in storage could be caused by non-enzymatic browning reactions, e.g., lipid peroxidation [19], and, at the same time, could also be influenced by the decrease in moisture content and the resulting compacting of the samples. We hypothesize that this might be the reason for the color trend observed between the Saanen samples during the storage days.

### 3.2. Characterization of the Fatty Acid Profile

The fatty acid profile of ricotta T0 is reported in Table 2. The main differences were the higher percentage of stearic (C18:0, *p* < 0.05), vaccenic (C18:1 trans-11, *p* < 0.05) acids, and conjugated linoleic acids (CLA, *p* < 0.01) in the Teramana breed. Furthermore, it was possible to observe a greater presence of myristic (C14:0, *p* < 0.01), palmitic (C16:0, *p* < 0.01), and palmitoleic (C16:1, *p* < 0.01) acids in the Saanen breed. Furthermore, higher desaturation indexes of C14:0 and CLA were observed in Teramana Ricotta samples.

Teramana ricotta samples were characterized by a lower content of myristic (C14:0) and palmitic (C16:0) acids. For the myristic acid, this result might be mainly related to higher expression levels or simply higher activity levels of the enzyme Δ9-desaturation (SCD) implicated [20]. This result is supported by the increase in the C14:1/C14:0 ratio, which Mele et al. [21] considered a reliable index of SCD in the mammary gland.

Regarding palmitic acid, it could be hypothesized that the reduced levels of C16:0 in these samples result from the activity of elongase-6, which more frequently converts C16:0 into C18:0 (elongase of the long-chain fatty acid family 6, ELOVL6). This microsomal enzyme catalyzes the elongation of the palmitate stearate chain and is present ubiquitously in almost all mammalian tissues, thereby influencing the fatty acid composition of tissues. [22]. Several studies have demonstrated the higher expression or activity of some enzymes in rustic breeds [23,24]. Indeed, in Teramana samples, we found a higher relative percentage of stearic acid (C18:0).

Another hypothesis that would lead to an increase in stearic acid is that of rumen biohydrogenation. Indeed, the increased presence of vaccenic acid (C18:1 trans-11) could promote increased conversion leading to the formation of stearic acid. At the same time enzyme activity produces higher CLA, confirmed also by a higher CLA desaturation index. Ruminant products are the main dietary source of CLA. Because of the many health advantages for consumers, an increase in these FA in cheese is thought to be a good thing. The immune system is modulated by CLA, which also helps to prevent the build-up of body fat and slows the progression of atherosclerosis [25].

### 3.3. Protein Profile of Ricotta

The SDS-PAGE analysis was used to characterize the protein profile of ricotta (Figure 1). SDS-PAGE analysis of ricotta T0 and T5 samples showed the separation of the main whey protein fraction (lactoferrin, serum albumin, immunoglobulin, β-lactoglobulin, and α-lactalbumin) and less intensive bands corresponding to casein residues.

In Teramana ricotta, a higher intensity of casein residues (*p* < 0.001) and β-lactoglobulin (*p* < 0.001) were observed compared to Saanen ricotta samples. The opposite trend was observed for α-lactalbumin (*p* < 0.01), which was higher in Saanen ricotta samples (Table 3). No statistical differences were observed for the conservation.

SDS-PAGE analysis showed significant differences in the main protein fractions, casein residues, β-lactoglobulin, and α-lactoalbumin. The strength of binding and affinity between proteins may be affected by several factors, such as pH and ionic strength [26].

In the Teramana samples, we found a higher relative concentration of β-lactoglobulin and casein residues and a lower relative concentration of α-lactalbumin concerning the Saanen samples. Interestingly, the relative abundance of β-lactoglobulin in the Teramana samples confers the properties of functional food, particularly the biological activities of β-lactoglobulin including antiviral, pathogen adhesion prevention, and hypocholesterolemic effects [27].

It is noteworthy to mention that thermal treatment to produce ricotta cheese leads to the unfolding of globular protein structures and the formation of new protein–protein interactions. At 25 °C and neutral pH, β-lactoglobulin is predominantly present as a dimer in solutions. However, when the temperature is raised to 87–89 °C for ricotta cheese production, various conformational changes occur, one of which is the exposure of the thiol groups [7,28,29]. It appears logical to deduce that the differences observed in relative abundances among whey proteins between the Teramana and Saanen samples are connected to a modification in the association behavior of β-lactoglobulin, casein residues, and α-lactalbumin; further analysis will be needed to clarify this aspect.

### 3.4. Evaluation of Volatile Compounds

Thirteen VOCs were identified in T5 ricotta samples belonging to alcohols, esters, ketones, and carboxylic acids families (Figure 2). Hexadecanoic acid and butyric acid were higher in Teramana ricotta cheese (*p* < 0.05); on the contrary, a lower relative percentage was observed in ketones (2-heptanone and 2-nonanone) and carboxylic acids (propanedioc acid and n-decanoic acid) (*p* < 0.001).

The investigation of the volatile profile (VOCs) of ricotta cheese samples allowed us to identify four types of chemicals, primarily originating from the lipolytic process: alcohols, carboxylic acids, ketones, and esters. Many of the compounds identified in both samples belong to carboxylic acids produced by the breakdown of triglycerides by the enzymes of microbial, milk, and rennet origins [30]. Carboxylic acids directly contribute to the development of cheese flavor, and, on the other hand, they indirectly contribute to the development of methyl ketones, secondary alcohols, aldehydes, lactones, and esters [31]. The elevated production of these compounds could likely be attributed to an increase in the lipolysis of triglycerides by microbial and endogenous milk enzymes, leading to a heightened release of FFAs [32]. Furthermore, this could be explained by the extent of the autolysis of starter cells, a phenomenon driven by the increased activity of various peptidoglycan hydrolases, commonly referred to as autolysins. These enzymes are characterized by a C-terminal domain containing a zinc-binding motif [33], which serves as an important cofactor in the mechanisms leading to bacterial cell lysis. This process results in the release of peptidases and, notably, lipases that accelerate the lipolytic event in cheese [30].

The samples from Teramana showed lower levels of free fatty acids. These substances are frequently linked to potent and disagreeable scents that have been described as sweaty, rancid, and cheesy [32]; as a result, their excessive production may have negative impacts on the assessment of the flavor and aroma of dairy products. These carboxylic acids can be oxidized to produce β-ketoacids, which can then be swiftly decarboxylated to produce the equivalent methyl ketones.

The higher percentage of ketones in Saanen ricotta cheeses could depend on a greater presence of carboxylic acids, as observed in the present study. According to reports, methyl ketones, which give dairy products their distinctive flavor, are described as fruity, peppery, and musty [34]. 2-heptanone and 2-nonanone were shown to be greater in Saanen ricotta cheese. The biosynthesis of these two methyl ketones is mostly linked to mold metabolism. In addition to a common fruity flavor, 2-nonanone has tea-like/medicinal/sour notes, and 2-heptanone has a blue cheese/mushroom flavor [35].

On the other hand, the greatest number of esters in Teramana samples could be the result of higher esterification of alcohol and carboxylic acids during storage. This could have a positive impact on the sensorial properties of the ricotta cheese as the ethyl esters are generally responsible for pleasant, fruity notes that reduce cheese sharpness and bitterness.

## 4. Conclusions

In conclusion, this study provides significant evidence for the enhancement of endangered goat breeds, such as Teramana goats. The results indicate that dairy products derived from this local breed, rich in CLA, exhibit noteworthy advantages. Furthermore, Teramana samples exhibited reduced levels of ketones and carboxylic acids, indicating greater oxidative stability of Teramana goat ricotta cheeses during storage compared to Saanen samples. Additionally, Teramana samples showed no changes in color indices during storage, unlike Saanen goat ricotta, where samples were darker at the end of storage. Promoting the production and consumption of such products would not only contribute to the preservation of genetic and cultural diversity but also yield economic benefits for local communities engaged in their production. Nonetheless, further research is necessary to deepen the understanding of the unique characteristics of this breed and to develop effective strategies for its long-term protection and enhancement.

## Figures and Tables

**Figure 1 foods-13-01239-f001:**
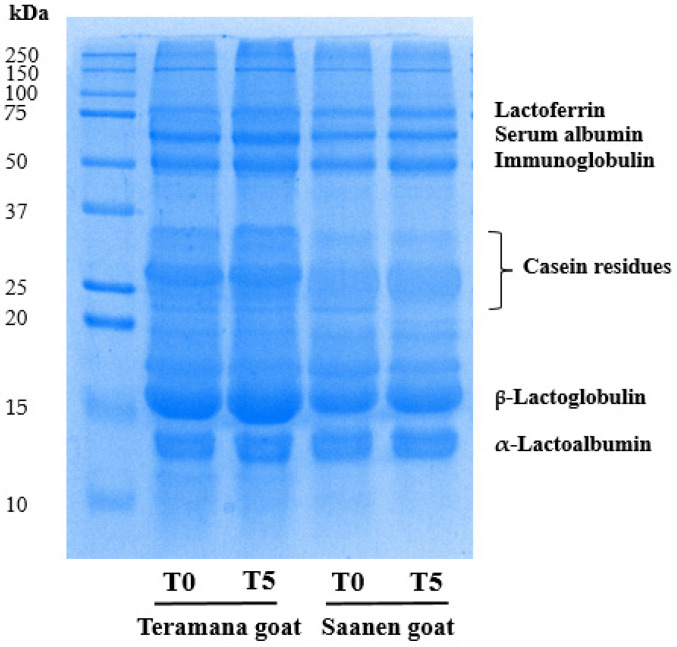
A representative image of SDS-PAGE analysis of ricotta at 0 (T0) and 5 (T5) days of ripening, obtained from the milk of Teramana and Saanen goats. (n = 3).

**Figure 2 foods-13-01239-f002:**
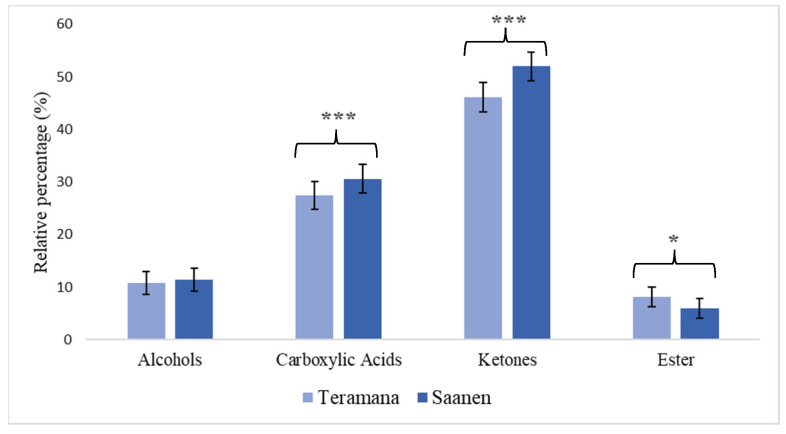
The main classes of volatile compounds detected in ricotta cheese at 5 (T5) days of conservation were obtained from the milk of Teramana and Saanen goats. All data are reported as least square means ± standard error. * *p* < 0.05; *** *p* < 0.001.

**Table 1 foods-13-01239-t001:** The physical and chemical properties of ricotta at 0 (T0) and 5 (T5) days of ripening were obtained from the milk of goat breeds: Teramana and Saanen.

	T0		T5			*p*-Value		
	Teramana	Saanen	Teramana	Saanen	SEM	R	T	R × T
Chemical composition (%)								
Dry Matter (DM)	35.23 b	29.36 d	38.35 a	31.39 c	0.47	0.001	0.001	ns
Total lipids	27.96 b	19.37 c	33.47 a	20.06 c	2.13	0.001	0.05	ns
Color								
L*	94.90 a	94.97a	94.84 a	93.69 b	0.21	ns	0.01	ns
b*	5.67 b	5.69 b	5.74 b	6.57 a	0.04	ns	0.001	ns
YI	8.54 b	8.56 b	8.64 b	10.03 a	0.12	ns	0.001	ns
ΔE	0.42		1.42					

All data are reported as least square means. L*: lightness; b*: yellow/blue chromaticity; YI: 142.86. (b*/L*). Humidity is reported on a dry matter basis (DM). Different letters in the same row indicate significant differences. SEM: pooled standard error of the mean. R: breed; T: conservation; ns: not significant.

**Table 2 foods-13-01239-t002:** The fatty acid composition of ricotta cheese (T0) was obtained from two goat breeds: Teramana and Saanen.

	Teramana	Saanen	SEM	*p*-Value
C4:0	2.61	1.84	0.78	ns
C6:0	3.04	2.35	0.99	ns
C8:0	3.76	3.15	1.47	ns
C10:0	12.30	12.25	0.63	ns
C12:0	5.91	4.17	0.76	ns
C14:0	9.59 b	12.46 a	0.46	0.01
C15:0	0.98	1.13	0.01	ns
C16:0	23.65 b	25.90 a	1.79	0.01
C18:0	13.91 a	11.89 b	1.42	0.05
C20:0	0.36	0.19	0.01	ns
C14:1	0.43	0.43	0.01	ns
C16:1	1.11	1.00	0.61	ns
C18:1, t11	1.81 a	1.47 b	0.19	0.05
C18:1, c9	15.77	14.18	1.93	ns
C18:1, c11	0.14	0.32	0.01	ns
C18:2	1.79	1.50	0.08	ns
C18:3	1.55	1.55	0.06	ns
SFA	74.43	77.12	1.16	ns
MUFA	18.77	17.42	1.22	ns
PUFA	3.34	3.35	0.30	ns
CLA	1.86 a	1.11 b	0.06	0.01
OTHERS	1.57	1.62	0.01	ns
DI C14	4.32 a	3.40 b	0.15	0.05
DI C16	2.84	3.04	0.32	ns
DI C18	53.05	52.64	0.49	ns
DI CLA	48.50 a	44.58 b	1.71	0.01

All data are reported as least square means percentage of total fatty acids. Different letters in the same row indicate significant differences. SEM: pooled standard error of the mean; ns: not significant. CLA = conjugated linoleic acids; SFA = saturated fatty acids; MUFA = monounsaturated fatty acids; PUFA = polyunsaturated fatty acids; DI = desaturation index.

**Table 3 foods-13-01239-t003:** Densitometric analysis of SDS-PAGE protein bands (Figure 1) of ricotta at 0 (T0) and 5 (T5) days of conservation, obtained from the milk of the goat breeds Teramana and Saanen.

	T0		T5			*p*-Value		
	Teramana	Saanen	Teramana	Saanen	SEM	R	T	R × T
Lactoferrin	4.82	5.50	5.01	5.45	1.36	ns	ns	ns
Serum albumin	9.19	9.22	8.31	9.16	0.13	ns	ns	ns
Immunoglobulin	15.10	15.37	14.80	14.41	0.17	ns	ns	ns
Casein residues	24.09 a	19.59 b	23.93 a	20.76 b	2.15	0.001	ns	ns
β-lactoglobulin	29.25 a	24.72 b	30.94 a	25.15 b	1.60	0.001	ns	ns
α-lactoalbumin	17.52 b	25.56 a	16.97 b	25.03 a	2.30	0.01	ns	ns

All data are reported as relative least square means; the percentages of the total protein content are shown in the reference column. Different letters in the same row indicate significant differences. SEM: pooled standard error of the mean. R: breed; T: conservation; ns: not significant.

## Data Availability

The original contributions presented in the study are included in the article, further inquiries can be directed to the corresponding author.

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
