# Peer review of "Evaluation of Chemical and Nutritional Characteristics of Ricotta Cheese from Two Different Breeds: The Endangered Italian Teramana Goat and the Cosmopolitan Saneen Goat"

_foods, 2024, doi:10.3390/foods13081239_

Round 1

Reviewer 1 Report

Comments and Suggestions for Authors

Abstract:

Line#16: storage may be an appropriate word in the place of conservation

Authors' recommending promoting indigenous breeds goes beyond the scope and findings of the study. So, it would be a good idea to keep the conclusion statements in consistent and limited to the study scope

Introduction:

The first few paragraphs are talking about the native breeds population decline in relation to the other commercial type. Whereas the scope of study was the advantages of a particular product make out of two different breeds, so more discussion about the qualitative difference between the produce from each type of the goat studies would be more meaningful.

Similarly, the discussion of cheese whey and cheese make process at this length is not at all relevant to the study. It would be more valuable if the nutritional quality is compared and contrasted here.

Comments on the Quality of English Language

Overall, a well written manuscript, however, a minor English language verification would improve further the readability of the manuscript.

Author Response

The Authors thank the Editor and Referees for their careful and rapid review of our manuscript. All the modifications suggested by the Referees have been appreciated and followed, and we consider that the manuscript has considerably improved now. Please find enclosed a clean version of the revised manuscript (foods-clean_version) and a file with all the highlighted modifications (foods_highlighted).

We have extensively revised the manuscript according to the attached iThenticate report and we really hope that you could reconsider our new version of the manuscript suitable for publication in Foods. We remain at your disposal in case you need any other clarification or additional information.

Looking forward to hearing from you, we send you our best regards.

Reviewer 2 Report

Comments and Suggestions for Authors

The manuscript “Evaluation of Chemical and Nutritional Characteristics of Ricotta Cheese from an endangered breed of Italian goat Teramana Goat” is a significant research endeavor. It aims to evaluate the characteristics of ricotta from milk derived from the Teramana goat, comparing it with that produced by the Saanen goat, a more common breed. The originality of this research is not just in the comparison but also in the promotion of dairy products originating from milk from indigenous breeds. This promotion has significant future impacts on biodiversity conservation, making this study a crucial contribution to the field.

The manuscript is well structured, clearly describing the goals, methods, and results. However, some results would benefit from a deeper discussion. A few issues on the manuscript are listed below:

Page 2, Line 67 “(…) in favor of saturated fatty acids and 67 CLA (linoleic acid conjugates) features connected to consumer welfare.” There seems to be a confusion here. It appears that the authors might have intended to refer to 'unsaturated' fatty acids instead of 'saturated '. Could you please confirm if this is the case?

P3L112. Why the fatty acid profile was only determined in T0 samples? Couldn't it change after storage?

P4L157. “(…) are reported to produce high-quality animal products.” Please explain it better. The same inference was made in the manuscript hypothesis (introduction section); however, no additional information was given to support it. Does " quality " refer to the nutritional composition and sensory profile? Why would Terramana milk yield better-quality products when compared to products from other “common” breeds?

P4L187. Please explain the factors involved in the occurrence of Maillard reaction in fresh cheese stored for only five days at low temperatures. Is there any other research showing Maillard products so early in cheese?

P5L199. Although the authors mention surveying goat milk producers on the animals' diets, diet differences/similarities between the two studied breeds are not discussed in section “3.2. Characterization of the Fatty Acid Profile.” Couldn't the feed composition influence the observed differences?

P5L223. “Ruminant products are the 223 main dietary source of CLA.” The phrase is repeated.

Table 3. Please correct: β-lactoglobulin instead of β-lattoglobulin

Author Response

(The authors gave the same response as above.)

Reviewer 3 Report

Comments and Suggestions for Authors

Comments

In this work entitled “Evaluation of Chemical and Nutritional Characteristics of Ri- 2 cotta Cheese from an endangered breed of Italian goat, Tera- 3 mana Goat”, authors evaluated the chemical and nutritional characteristics of two different breed. There are some problems 

1. The abstract is relatively simple and important results are not well described. It is hard to attract attention from readers. It is difficult for readers understand the novelty of the study.

2. The authors described that “In order to assess the impact of different breeds on qualitative characteristics of ricotta cheese, various steps were taken to analyze its total lipid content, fatty acid composition, volatile profile, and protein profile.” So, the title of the manuscript should be corrected to better reflect authors aim.

3. Line 76-77, should be deleted, because it is background of authors scientific project not the background of the present study.

4. For fatty acid profile and volatile substances analysis, I think the design is wrong. T0 and T5 should be evaluated simultaneously to better understand the changes and (or) development of fatty acid and volatile substances. More results and discussion on volatile flavor substance should be described.

Author Response

(The authors gave the same response as above.)

Round 2

Reviewer 2 Report

Comments and Suggestions for Authors

The manuscript titled “Evaluation of Chemical and Nutritional Characteristics of Ricotta Cheese from an endangered breed of Italian goat Teramana Goat” was revised accordingly. No relevant issues were detected in the second version.

Reviewer 3 Report

Comments and Suggestions for Authors

The manuscript was carefully revised according to my comment.